# Masking is good, but conforming is better: The consequences of masking non-conformity within the college classroom

**Jessica Sullivan**[1]*, **Corinne Moss-Racusin**[1], **Kengthsagn Louis**[2]

**1** Skidmore College, Saratoga Springs, New York, United States of America, **2** Boston College, Chestnut Hill, Massachusetts, United States of America

* jsulliv1@skidmore.edu

**Data Availability Statement:** All relevant data for this study are publicly available from the OSF repository (https://osf.io/u7b28/).

## Abstract

In the years following the acute COVID-19 crisis, facemask mandates became increasingly rare, rendering masking a highly visible personal choice. Across three studies conducted in the U.S. in 2022 and 2023 ($N$ = 2,973), the current work provided a novel exploration of the potential impacts of adhering to vs. deviating from group masking norms within college classrooms. Experiments 1 and 2 used causal methods to assess the impact of hypothetical target students' masking behavior on participants' beliefs about that student's classroom fit (e.g., how well they fit in, how much their professor likes them, whether they are invited to study group). Maskers were expected to experience more classroom inclusion relative to non-maskers, but the largest effects were conformity effects: participants expected that students who deviated from a class's dominant mask-wearing behavior would experience massively lower classroom fit. Study 3 used correlational and qualitative methods to establish the real-world impact of mask conformity in a diverse sample of college students. Students reported believing that masking–and mask conformity–impacted others' perceptions of them, and reported avoiding deviating from masking norms. Students reported that their desire for mask-conformity impacted both their willingness to enroll in courses and their actual masking behavior, suggesting both academic and public health impacts. Across all three studies, we asked whether pressures to conform have disproportionate effects on particular groups, by exploring the effects of gender (Studies 1 and 3), immune-status (Studies 2 and 3) and race (Study 3). Our data raise important issues that should be considered when determining whether to e.g., enact mask mandates within college classrooms and beyond, and for understanding the cognitive and social consequences of mask wearing.

## Introduction

> *"I always worried if other students would be wearing a mask or not. There were some days I would wait for the professor to put on their mask or ask the students to wear masks before I put on mine"*

> *–Free Response, Study 3, Participant 168*

**Funding:** Funding from Skidmore College was awarded to JS in support of this project. The funders had no role in study design, data collection and analysis, decision to publish, or preparation of the manuscript.

**Competing interests:** The authors have declared that no competing interests exist.

In response to the COVID-19 pandemic, communities and organizations imposed–and over time, relaxed–mandates requiring individuals to wear face masks [1, 2]. While the benefits of face-masking for preventing airborne disease-spread are well established [3], the choice to mask is now generally an individual one within the U.S. Unlike many individual health-related decisions which can remain private, mask-wearing is highly visible. This means that masking could serve as an important cue in shaping social judgments and person perception [4]. This paper investigates how masking behavior impacts social judgments by–and about–college students.

## How might face-masking impact perceptions?

We considered three primary possibilities for how masking behavior might shape social perceptions. One possibility is that masking behavior (i.e., wearing a mask or not) is itself predictive of social judgments [5]. That is, people who routinely wear masks may be perceived differently relative to those who do not. For example, some work has found evidence of positive views of mask-wearers, in that they were perceived as more competent [6], physically attractive [7] and trustworthy [8] than comparable unmasked targets. Other work finds evidence of more negative perceptions of mask-wearers, in that they were seen as more psychologically distant [9] and *less* trustworthy [10]. Masking behaviors may not only impact the valence of evaluations, but could also be perceived as signaling additional characteristics such as an individual's risk aversion, political orientation, and toughness [11]. This idea aligns with substantial social psychological research from outside the study of face-masking behavior, revealing that both choices in attire [12, 13] and health-related decisions [14] can lead to highly valenced perceptions. By this view, one might predict that masking behavior itself could predict social and educational judgments in consistent ways.

A second possibility is that the effect of masking on social perceptions could interact with characteristics of the mask-wearer (e.g., race [15]) and/or the perceiver (e.g., political orientation [16]). Critically, and consistent with data from earlier during the COVID-19 pandemic, rates of masking continue to differ across groups: women were more likely to mask than men, 18–29 year-olds were more likely to mask than their older counterparts, and people of color were more likely to mask than White people [17]. Because people of color, immunocompromised individuals, and men are especially likely to be negatively impacted by COVID-19 (and the former two groups are also especially likely to mask; see [18] for review), understanding whether the impact of masking behavior on perceptions is related to e.g., gender, race, and immune-status is critical.

Of importance, previous work from outside the study of face-masking reveals that individuals encounter backlash (i.e., social and economic penalties) when they violate stereotypic expectations about how people from their group should behave [19]. For example, women who enact the male gender-typed trait of dominance are viewed as less likable and are less likely to be promoted relative to identically-dominant men [20], while men who enact the female gender-typed trait of modesty face similar consequences [21]. In the context of masking, perhaps visibly displaying the need for protection (e.g., by masking) could result in perceived stereotype-violations (and associated penalties) for men. Further data support the idea that masking decisions may be perceived differently depending on people's intersecting group memberships. For example, Black men wearing face masks are perceived more negatively (i.e., as more threatening and less trustworthy) than those without masks, while White men are perceived positively for wearing masks [22]. And, minoritized racial groups within the U.S. reported higher perceived penalties for masking [23]. These studies suggest the importance of understanding whether the impact of masking behaviors on social judgments also interacts

with group membership (as it does for other choices of attire; see [13] for review, such that masking behavior impacts perceptions of individuals from some groups more (or in different ways) than other groups.

Another major possibility—not mutually exclusive with the previous two—is that individuals are judged on whether their mask-wearing behavior conforms to that of their peers. This idea is consistent with classic work in social psychology highlighting the prevalence of conformity pressures on behavior [24] and the importance of norm-setting and adherence [25]. For example, conforming to established clothing trends (whatever those specific trends may be) is associated with increased peer acceptance among adolescents [26]. Similarly, adolescents' use of tobacco [27] and illegal drugs [28] is directly related to pressures to conform to the perceived usage patterns of their peers. Further, substantial literature–from research on minimal groups to the use of clothing to indicate group affiliation (see [29] for review)–has shown that even seemingly arbitrary choices in attire can signal group membership, leading to different social judgments for those who conform to vs. deviate from dominant behavior. Of course, masking is *not* an arbitrary clothing choice, but rather one that may be motivated by beliefs, values, and health concerns. Given the potential potency of masking as both a cue to group membership and as a signal about the individual, we asked whether conforming to–or deviating from–the dominant mask-wearing behavior within a classroom changes perceptions of social belonging within the class, regardless of whether that behavior is best for individuals' unique health situations.

Supporting the possibility of detecting conformity effects, while there has been substantial variability in mask-wearing behavior across place and time, rates of mask-wearing during the acute phase of the COVID-19 pandemic correlated among individuals who were in close physical proximity with one another [30]. Further, some studies have found that individuals report feeling more willing to mask when others around them are also masking [31], and that the belief that most people wear face-masks is positively correlated with masking behavior [32]. Further, while discrimination against mask-wearers fluctuated throughout the acute phase of the COVID-19 pandemic, individuals who wore masks in U.S. states without mask mandates reported high levels of COVID-19-associated discrimination, as did individuals who went unmasked in states with mask mandates [33]. These data raise the possibility that individuals experience penalties when they deviate from the dominant mask-wearing behavior–whatever that behavior may be. In other words, consequences may be associated largely with masking conformity vs. deviance, rather than masking vs. not.

To differentiate these possibilities, across 3 studies, we studied the potential effects of (a) masking (in general); (b) masking (in interaction with group membership); and (c) masking conformity. If masking behavior leads to social, academic, or professional penalties—for all groups, for some groups, and/or as a function of group norms—then researchers must examine how policies that allow for "individual choice" in behavior may act as systemic barriers to education/inclusion. For this reason, we asked whether masking behavior interacts with gender (Studies 1 and 2), immune-status (Studies 2 and 3), and other social group membership (Study 3).

## Masking in the college classroom

As a starting place for this line of research, we focused all three of our studies on the college classroom context. We did this because classrooms are places where social judgments can have both academic/professional and interpersonal repercussions, and where social group membership can greatly impact a student's perceived belonging within the classroom space [34]. Across all three studies, we pay special attention to perceptions of classroom fit—i.e. a metric

of how much an individual is perceived to be included, belong in, and fit with their classmates. Perceived belonging within the classroom is both personally and professionally important [35, 36], and may impact access to critical resources for success [37, 38]. Also, college classrooms are confined spaces (thus being potential locations for airborne disease spread) where the same group of individuals are present on a routine basis (thus allowing for the meaningful formation of social groups and dominant norms within the classroom). This renders them pertinent settings for understanding the relationship between social perceptions and engagement with crucial disease-mitigating strategies. Finally, at an applied level, the college classroom is an especially promising setting: professors and college administrators may have greater latitude in shaping the masking policy within the classroom than individuals in other sectors. Thus, we considered this environment ideal for exploring whether mask-wearing impacts perceptions of fit within the classroom, and whether this might be amplified by other social cues to group membership.

To summarize, in the present manuscript, we studied participants' social judgments of hypothetical target students as a function of targets' masking behavior (Studies 1 and 2), and also assessed students' self-reported experiences of the impact of masking behaviors on their own time in the college classroom (Study 3). The current work took a psychological approach, and focused primarily on understanding perceptions of and beliefs about masking behavior. Why focus on perceptions and beliefs? There is substantial theoretical reason to think that perceptions and beliefs about behaviors are an important psychological outcome in their own right, especially when predicting health-related outcomes. Indeed, perceived social consequences are known to shape social behavior and professional choices [39], and important health-related decisions (e.g., condom use, risky behavior) are known to be impacted by beliefs about how those decisions will be perceived by others [14].

By this logic, if students perceive that they (or others) are likely to be negatively evaluated for wearing (or not wearing) masks in a certain classroom, the mere belief that this outcome is likely–even absent first-hand experiences of it–may shape future behaviors. Indeed, we suspected that individuals might consider *self presentation* [40]—i.e., how they want to present themselves to others—when making evaluations related to masking behavior. Recent research has shown that people's beliefs about how others' impressions of them would be impacted if they wore a mask were negatively correlated with their intentions to wear a mask [32]. And, self presentation concerns have been robustly demonstrated to impact a variety of public health decisions [14, 41]. To this end, we explicitly study the role of self presentation concerns in shaping judgments about masking in Studies 2 and 3 (see Tables 1 and 2). Of course, perceptions and beliefs can sometimes be poorly calibrated to real world experiences, and so we also asked participants to report their own experiences of other's responses to their masking choices (Study 3), in order to provide evidence that real-world masking behavior can be associated with the types of social and academic consequences we experimentally assessed (Studies 1 and 2).

## Overview of research

We conducted two experiments (Study 1 and Study 2) and one self-report study (Study 3). All three were designed to explicitly test the perceived impact of masking and mask conformity on college students, as well as to explore how group membership (e.g., gender, immune-status, race) interacted with masking behavior in shaping perceptions. However, there were important methodological differences across each study, which we summarize in Table 1 and describe below.

**Table 1. Overview of methods for studies 1–3.**

| | Study 1 | Study 2 | Study 3 |
|---|---|---|---|
| Methodological Approach | | | |
| Experimental methods | X | X | |
| Qualitative and correlational methods | | | X |
| Masking Effects | | | |
| Tested for overall effects of masking | X | X | X |
| Tested for effects of masking differentially by group | X | X | X |
| Tested for effects of mask-conformity | X | X | X |
| Dependent Variables | | | |
| DV includes perceived classroom fit | X | X | X |
| DV includes public-health relevant measures (e.g., self presentation) | | X | X |
| Group Membership | | | |
| Tested for effects of gender | X | | X |
| Tested for effects of immune-status | | X | X |
| Tested for effects of race | | | X |

For the two experiments (Studies 1 and 2), we asked adult lay participants (recruited online via CloudResearch–formerly TurkPrime [42]) to report their beliefs about how target students' masking choices might impact those students' experiences of classroom fit. For those experiments, we recruited a broad sample of US adults (instead of, for example, restricting our participant pool only to students). The third study contains self-reports from college students who attended college during and immediately after the acute phase of the COVID-19 pandemic about the impact of masking behavior on their own experiences in the classroom. These students were recruited online using CloudResearch and SONA, and so the participant pool was not restricted to just one university or college.

As can be seen in Table 1, each of our studies was designed to measure the potential impact of group membership on judgments. In addition, because this research line was iterative, we added several public-health related measures in Studies 2 and 3, in order to more closely align our research questions to the health psychology literature, with a special focus on understanding the role of self presentation concerns in shaping judgments. To clarify and theoretically motivate the choices that were unique to each study, we provide a brief introduction and literature review prior to each study.

## Data collection

Data were collected between August, 2022 and May, 2023, with specific dates of data collection described below. Data were collected during a time when relatively few (if any) mandatory mask mandates were in effect on college campuses, and during a time when there were no statewide masking mandates within the U.S. Beginning in August, 2022, the CDC guidance only explicitly recommending masking after a known COVID-19 exposure or during an active infection [1]. Thus, mask wearing behavior during this period of time was not mandated, and unmasking did not indicate failure to comply with local laws. While masking has become less common in the U.S. since the beginning of the COVID-19 pandemic, by some measures, rates of masking remained relatively consistent across the time period during which we collected our data. In August, 2022–the start of data collection for Study 1–54% of U.S. individuals reported wearing a facemask at least once in the previous week [43]. As of May, 2023–the end of data collection for Study 3–46% of U.S. adults reported wearing a facemask during the previous week [17].

## Preregistration

For each study, data collection procedures occurred as preregistered (with all minor deviations noted in the main text and/or Supplemental Online Materials). All materials and preregistrations are available on OSF (https://osf.io/u7b28/). As a result of a clerical error, the preregistration for Study 3 was never formally submitted. However, a PDF of the preregistration, along with timestamped documentation of when it was last edited (i.e. prior to data collection) is available under the Study 3 materials on OSF [44].

All data exclusions, manipulations/measures, and analyses are reported in full in either the main text or in the S1 Appendix.

## Study 1

*"I tend to put on a mask when others are wearing a mask and I tend to not wear one when others are not wearing them. I think that I tend to conform to whatever everyone else is doing to protect themselves from the virus."*

–Free Response, Study 3, Participant 97

Study 1 used experimental methods to test two primary questions: are maskers perceived to have better classroom fit (i.e., is there a main effect of masking) relative to non-maskers, and are mask-conformers perceived to have better classroom fit than mask-deviants (i.e., is there a main effect of conformity)? In addition, Study 1 asked whether gender interacted with mask-wearing in shaping perceptions.

We studied gender for two main reasons. First, prior research suggests that gender may be associated with mask-wearing behavior, with women being typically more likely to mask than men [18, 45–48, but see 49]. We also know that *perceptions* of mask-wearing may be related to gender. Recent work has found that endorsement of masculine norms surrounding "toughness" predicts negative attitudes about mask-wearing [11, 50]. Similarly, mask-wearing may amplify or interact with existing gender stereotypes. For example, masking may reinforce stereotypes that women are more communal, more likely to take on a caregiving role, are in greater need of protection, are more fearful, etc. Or, masking may violate stereotypes, e.g., by showing that men are also sometimes in need of protection. More generally, there is good reason to believe that health-related decision-making can be especially gendered [51]. This led us to ask whether the effect of masking on social and academic perceptions is shaped by the gender of targets and/or participants. We also explored whether masking impacted perceived gender stereotypes, and whether participant gender impacted judgments. Because gender effects are not the main focus of the present manuscript, these analyses are reported in full in the S1 Appendix.

### Study 1 Method

**Ethics statement.** All research for Studies 1–3 was conducted in compliance with Skidmore College's IRB, and written consent was secured online for each study.

**Testing context.** Data were collected online using Cloud Research [42] on August 4th, 2022 from participants located within the U.S.

**Participants.** A total of $N = 1590$ participants consented. Of these, $n = 265$ were excluded based on preregistered criteria ($n = 4$ failed to progress through the study, $n = 252$ failed to recall the main mask manipulation, $n = 9$ failed more than one attention check). This resulted in a final $N = 1325$. In order to obtain expedited IRB exempt status and facilitate recruitment,

we did not collect further demographic or potentially identifying information about participants.

## Materials and procedure

**Introductory phase.**   Participants were told that they would be asked to remember details about an individual ('B'). In all cases, B was described as a 20-year-old college student in an engineering class. Our study had a 3 (target gender) x 2 (target masking behavior) x 2 (class masking behavior) between-subjects design. Specifically, participants were told that (a) "**B is a man**", (b) "**B is a woman**", or (c) they were not given any gender information. Participants also learned that B "[**always/never**] wears a mask", and that "Most students in B's class [**always/never**] wear a mask". This design allowed us to examine reactions to two types of masking *conformers* (masked and unmasked), as well as two types of masking *deviants* (masked and unmasked).

**Inferences about target.**   Participants then completed $n$ = 15 questions about B, each on a four-point likert scale that intentionally lacked a midpoint. The scales were of the format "Definitely X", "Probably X", "Probably Y", "Definitely Y". Items were presented in random order. There were $n$ = 3 attention checks (B's age [X = 20; Y = 30], B's occupation [X = a banker; Y = a student], the class that B was taking [X = a literature course; Y = an engineering course]), and $n$ = 12 other items, described below.

The survey included one question about B's gender (X = a woman; Y = a man). This served as an attention check for the participant who had learned B's gender (i.e. to identify if they remembered the target's gender), and as a DV of interest for those who did not learn B's gender (i.e. to see if they inferred the gender of the target from the target's masking behavior).

**Main dependent measure.**   Our preregistered primary dependent measure was our "perceived classroom fit" scale, which contained three questions: Whether B fits in with their classmates (X = a misfit; Y = fits in), whether B was invited to the class's study group (X = not invited, Y = invited), and whether B's professor liked B (X = liked; Y = disliked). See Table 2 for full descriptions of scales used in Studies 1–3.

**Additional measures.**   We also measured $n$ = 8 other characteristics corresponding to gender stereotypes and stereotypes about mask-wearers (e.g., surrounding need for protection, independence, emotional strength, warmth, etc. . .); full reporting of these is available in the S1 Appendix.

**Memory checks.**   There were two memory checks: one asked about how often B wore a mask, and the other asked how often B's classmates wore masks. Participants responded by selecting one of three multiple choice responses: "Always", "Never", and "It didn't say". As noted above, only the $N$ = 1325 participants who provided correct responses to both of these questions (and met our other inclusion criteria) were included in our final analyses.

**Demographics.**   Finally, we asked participants' gender, and how often the participant currently wears a mask in public (Never/Rarely/Sometimes/Often/Always). We did not collect any other demographic data.

## Study 2 Results

**Analytic approach.**   All scales were centered around 0, such that 0 indicated either neutrality ('neither agree nor disagree') or the midpoint of behavior ('sometimes' masking). Positive scores indicated e.g., higher perceived classroom fit, while negative scores indicated lower perceived classroom fit. We report all analyses for our Perceived Classroom Fit scale in the main text as this was our preregistered primary DV; we report results for the additional DVs in the OSF and S1 Appendix, as we believe they will be useful for future researchers who wish

**Table 2. Scales used in Studies 1–3.** Example items for each scale are provided, and full scales and study materials are available on OSF.

| Study | Scale | Source | # items | Examples | α | Higher scores indicate |
|---|---|---|---|---|---|---|
| **Scales measuring impact of masking on the classroom** | | | | | | |
| Studies 1 and 2 | Classroom Fit | novel | 3 | Whether B fits in with their classmates (X = a misfit; Y = fits in), whether B was invited to the class's study group (X = not invited, Y = invited), and whether B's professor liked B (X = liked; Y = disliked) | $\alpha_{Study\ 1}$ = .79 $\alpha_{Study\ 2}$ = .71 | Greater fit and academic support |
| Study 3 | Mask Impact Scale | 3 questions modified from Classroom Fit Scale + 7 novel questions | 10 | "I feel like my professor will like me less if my masking behavior differs from that of my classmates"; "I feel like I have to choose between my safety and fitting in when it comes to masking." (1 "strongly disagree"—7 "strongly agree") | $\alpha_{Study\ 3}$ = .90 | Higher belief that masking decisions impacts social and academic outcomes |
| Study 3 | Classroom Choice | novel | 3 | "I am more likely to want to enroll in/stay in a class if I know that most students in class will always wear a mask"; "I am more likely to want to enroll in/stay in a class if I know that the professor will always wear a mask" (1 "strongly disagree"—7 "strongly agree") | $\alpha_{Study\ 3}$ = .795 | Larger impact on student's enrollment behavior depending on masking behavior. |
| **Health psychology scales** | | | | | | |
| Studies 2 and 3 | Self Presentation | Louis, et al. (2023) | 9 | Study 2: How hard does B work to.../How hard: "get others to respect them"; "give a good impression of their family"; "look/sound smart" (1 "does not work hard"—7 "works hard") Study 3: I think my masking behavior impacts... "whether others respect me"; "others' impression of my family"; "how smart I look/sound" (1 "strongly disagree"—7 "strongly agree") | $\alpha_{Study\ 2}$ = .89 $\alpha_{Study\ 3}$ = .81 | Greater effort to be perceived well |
| Study 2 | Social Acceptance | Keyes (1998) | 7 | B thinks that: "other people are unreliable"; "other people are kind"; "people live only for themselves" (1 "strongly disagree"—7 "strongly agree") | $\alpha_{Study\ 2}$ = .89 | More favorable views of and trust in others |
| Study 2 | Social Contribution | Keyes (1998) | 6 | B thinks that: "they have something valuable to give the world"; "their work provides an important product for society"; "their behavior has some impact on other people in the community" (1 "strongly disagree"—7 "strongly agree") | $\alpha_{Study\ 2}$ = .74 | Higher belief in one's own contribution to society |

to construct new theories about the impacts of gender and mask-wearing on perceptions of non-academic traits.

**Participant characteristics.** 1323/1325 participants reported their mask-wearing behavior, which was relatively evenly distributed (Never = 22.7%; Rarely = 21.8%; Sometimes = 22.7%; Often = 16.6%; Always = 16.2%); most participant self-identified as either male or female ($n_{male}$ = 643; $n_{female}$ = 656; $n_{nonbinary}$ = 12; $n_{noresponse}$ = 14).

**Simple gender effects.** Consistent with previous work suggesting that masking may be feminized [18, 45–48], women reported more frequent mask-wearing than men ($t$(1297) = -2.44, $p$ = .015, $d$ = .14 [$CI_d$ = .03-.24]; see S1 Appendix for a visualization, and for full reporting of effects relating to gender). In addition, when B's gender was not specified, participants were significantly more likely to identify B as a woman if B always masked than if B never masked ($t$(445) = 2.05, $p$ = .041, $d$ = .19 [$CI_d$ = .008-.38]), also suggesting that gender and mask-wearing behavior are perceived to be related to one another.

**Effects of mask-wearing on perceived classroom fit.** Our primary analysis predicted perceived Classroom Fit from B's mask-wearing behavior, the class's mask-wearing behavior, and their interaction. When B's gender was not specified, there were main effects of B's mask-wearing and the class's mask-wearing (see S1 Appendix for full reporting of this and all models, which were qualified by a significant interaction ($B$ = .25, $SE$ = .013, $p$ < .0001, $\eta_p^2$ = .43).

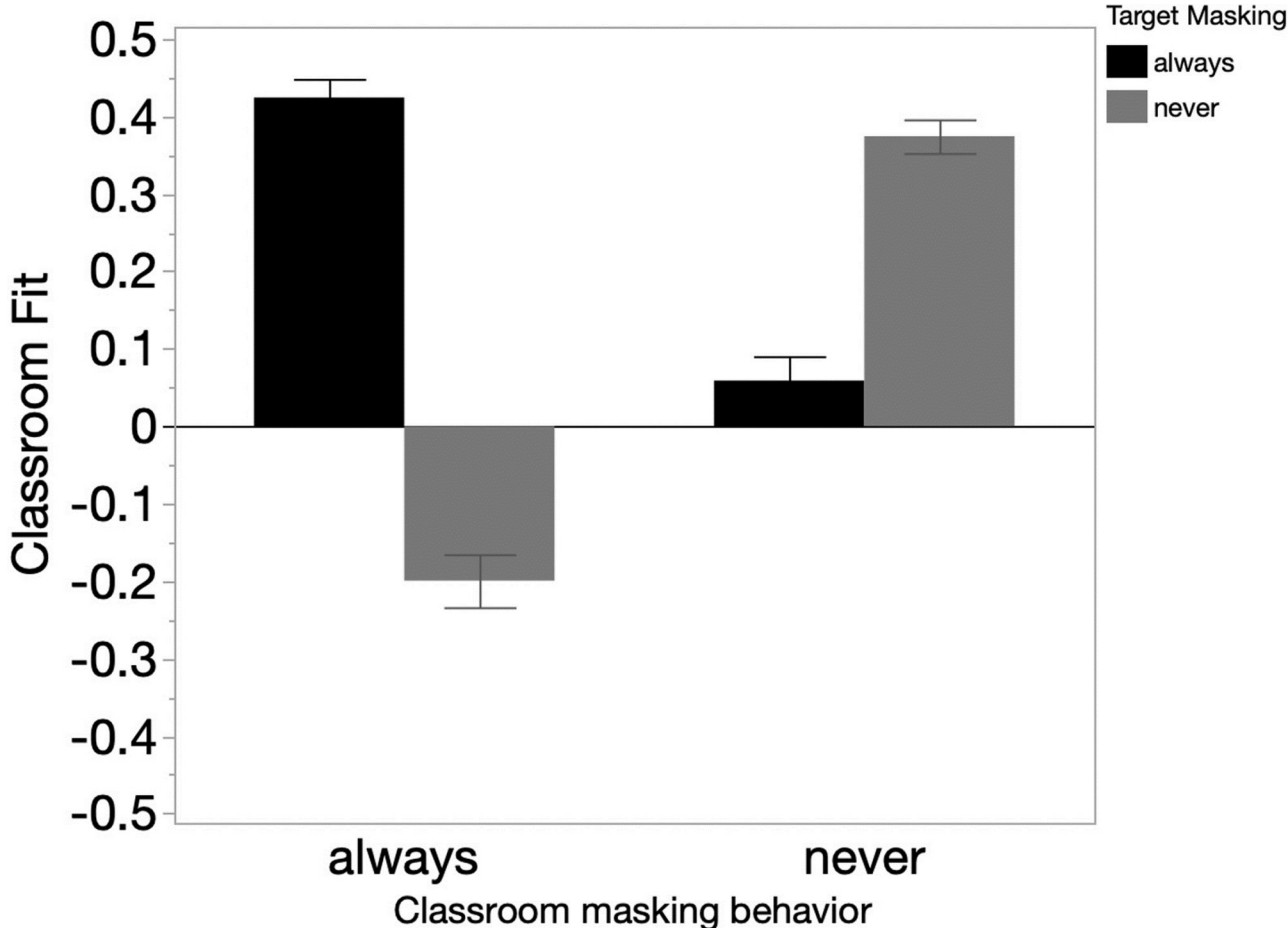

**Fig 1. Collapsing across target gender, perceived classroom fit (y-axis) by class masking behavior (x-axis), and target masking behavior (gray vs. black).** Error bars are 95% CIs.

Similarly, for participants who learned B's gender, we also included B's gender (and all interaction terms) in the model: for this analysis, we found a significant interaction of the class's mask-wearing behavior and B's mask-wearing behavior ($B = .23$, $SE = .009$, $p < .0001$, $\eta_p^2 = .45$); there were no effects associated with gender (Fig 1; S1 Appendix).

To better understand the interaction of target- and class-masking behaviors, we conducted a planned Tukey's HSD on the full dataset, collapsing across target gender. Both masked conformers (who *always* wore a mask when their classmates *always* did; $M = .42$) and unmasked conformers (who *never* wore a mask when their classmates also *never* did; $M = .37$) were perceived positively, and not differentially so from each other ($p = .08$, $d = .19$ [$CI_d = .03$-$.34$]). Critically, masked conformers were rated as having higher perceived classroom fit than both types of deviants (see Fig 1). Specifically, masked conformers ($M = .42$) were rated as fitting in significantly more than (a) masked deviants (who *always* wore a mask when their classmates *never* did, $M = .06$, $p < .0001$, $d = 1.38$ [$CI_d = 1.22$–$1.54$]) and than (b) unmasked deviants (who *never* wore a mask when their classmates *always* did, $M = -.199$, $p < .0001$, $d = 2.36$ [$CI_d = 2.18$–$2.53$]). Unmasked conformers ($M = .37$) were also rated as fitting in significantly better than both (a) unmasked deviants ($M = -.199$, $p < .0001$, $d = 2.17$ [$CI_d = 1.995$–$2.34$]), and than (b) masked deviants ($M = .06$, $p < .0001$, $d = 1.19$ [$CI_d = 1.03$–$1.35$]). Because masked

conformers did not receive different fit scores than unmasked conformers, these results suggest that mask-conformity itself predicted classroom fit irrespective of actual mask-wearing behavior (i.e., the third theoretical possibility discussed in our introduction). Indeed, these data provide novel evidence for an exceptionally strong conformity effect in the context of a highly important health-related behavior (Fig 1).

While our main finding was a conformity effect, not all deviants were perceived identically. Specifically, unmasked deviants (who *never* wore a mask when their classmates always did; $M$ = -.199) were rated as having significantly lower fit than masked deviants (who *always* wore a mask when their classmates *never* did; $M$ = .06, $p$ < .0001, $d$ = .98 [$CI_d$ = .82–1.31]). This is consistent with the belief that maskers have better fit than non-maskers, but only amongst masking deviants.

**Other effects of mask-wearing.** We conducted planned and structurally identical analyses to those reported above for the remaining 8 items that assessed the impact of masking on gender stereotypes. For 6 out of 8 items (all except "emotional strength" and "competence"), there were significant interactions of B's mask-wearing behavior and the class's mask-wearing behaviors, even when applying Bonferonni corrections for multiple comparisons (all $p$ < .0062). In other words, individual masking behaviors interacted with the behaviors of those in the surrounding classroom context when predicting social judgments; these effects were generally consistent with conformity effects, and generally did not interact with gender (see S1 Appendix for full reporting). These data suggest that mask-conformity may impact not only perceived classroom fit, but also perceptions of other characteristics. We return to this possibility in Study 2.

## Study 3 Discussion

Individuals whose mask-wearing behavior conformed with their peers'—relative to those whose behavior deviated—were perceived as being more likely to be invited to a study group, to fit in with their classmates, and to be liked by their professor. Further, both masked and unmasked conformers were rated highly (and equally) positively. While maskers were rated more favorably than non-maskers, mask-wearing behavior itself was a far weaker predictor of perceived Classroom Fit than was conformity to mask-wearing norms within the classroom. These data suggest that the dominant mask-wearing behavior within a classroom impacts the perceptions of individuals who fail to conform. While we found some contexts–reported in the S1 Appendix–in which masking behavior interacted with gender (either of the perceiver or the target) in shaping perceptions of individuals, conformity effects were much larger than gender effects and our data suggest that both men and women are judged for deviating from masking norms.

These results suggest that people believe that students will experience both interpersonal and academic penalties for failing to conform to the dominant masking behavior. This means that such conformity pressures could have a net positive effect on public health (e.g., in the context where the dominant behavior is to mask and disease transmission risk is high), but may also have a negative effect on public health (e.g., in the context where masking behavior is rare and disease risk is high). Given that individuals underestimate rates of mask-wearing among their peers [49, 52], the expectation that deviants will be perceived as fitting in worse with their classmates could lead to a feedback loop that promotes ever-lower rates of masking.

In addition, there may be health situations in which the safest behavior does *not* conform with the dominant masking behavior, as in classrooms where disease transmission is likely, but where most individuals remain unmasked and conformity pressures thus discourage masking. Immunocompromised individuals are at especially high risk for complications from

COVID-19 and other airborne disease, and may not feel the same freedom to go unmasked as do others in the classroom. In this context, students may be forced to choose between social acceptance and protecting themselves. For this reason, Study 2 addresses how the presence of immunocompromised individuals within the classroom impacts perceptions of masking behavior.

## Study 2

*"i* [sic] *always kind of assumed that the people that were wearing masks were immunocompromised or lived with someone that was."*

*- Free Response (Study 3, Participant 133)*

Approximately 2.8% of the U.S. population is immunocompromised [53] and therefore at higher risk of severe COVID-19 [54]. And, during the acute phase of the COVID-19 pandemic, immunocompromised individuals and their families experienced severe mental health consequences in response to the real threat that COVID-19 posed to them and their loved ones [55, 56]. Masking decreases the transmission of COVID-19 [57] and other airborne diseases, and can therefore protect immunocompromised individuals [18, 58]. Thus, immunocompromised people might wear facemasks even (or especially!) when others around them do not, as might individuals who wish to protect vulnerable groups or individuals. This raises a question: is an individual's mask-deviant behavior (i.e. failing to conform to the dominant masking behavior within the environment) also penalized when it is explicitly motivated by concerns related to immune status? Uncovering whether immunocompromised individuals are also expected to experience penalties for deviating from masking norms can help us understand the boundary conditions of mask-conformity effects and how to create supportive and safe classrooms for all students. More generally, the perceived consequences of being a masked deviant may be reduced in a context where individuals within the classroom are known to be immunocompromised, on the logic that e.g., being a masked deviant could be perceived as an act of solidarity.

As in Study 1, we asked how mask conformity predicted a target's perceived classroom fit. New to Study 2, we also measured the impact of classroom members' immune-status on perceived classroom fit, and on several additional outcomes that have been shown to be important to the health psychology literature. As discussed in the introduction, past research has shown that self-presentation concerns can prevent people from engaging in healthy behaviors, such as wearing condoms if they believe doing so could affect their social image [14, 41]. Of particular relevance to the COVID-19 pandemic, because masking could affect the mask wearer's social image, such self-presentation concerns might be particularly salient in the classroom where students seek belonging [59–61].

### Study 1 Method

**Testing context.**   Data were collected online on February 3rd, 2023 from participants located within the U.S.

**Participants.**   Participants (*N* = 2389) were recruited via Cloud Research Panels (formerly known as TurkPrime [42]); we excluded *n* = 924 based on preregistered criteria (e.g., failure to complete at least 80% of the study, failure on attention checks and/or memory checks, see S1 Appendix), resulting in a final *N* = 1465.

**Procedure.**   As in Study 1, participants were asked to remember details about a 20-year-old target ('B') who either "[**always/never**] wears a mask", and who was in a class where "Most

students in B's class [**always/never**] wear a mask". While in Study 1 we manipulated information about B's gender, in Study 2, we manipulated information about immune-statue. Specifically, participants were either not given any information about immune status (i.e. identical to Study 1), or were told that (a) B was immunocompromised; (b) B's classmate was immunocompromised; or (c) B's professor was immunocompromised. This allowed us to test whether perceptions of masking behavior were altered when it was assumed that B was protecting themself, a peer, or a high-status individual in the classroom (the professor).

As in Study 1, participants next answered questions about B on a four-point scale of the format "Definitely X", "Probably X", "Probably Y", "Definitely Y" (see Table 2 for scale information and OSF for full study materials). These questions included (a) two attention checks; (b) the Perceived Classroom Fit Scale; (c) a question about B's gender; (d) the Self-Presentation Scale; (e) the Social Acceptance Scale; and (f) the Social Contribution Scale.

**Demographics.** Finally, we asked participants' gender ($n_{male}$ = 513; $n_{female}$ = 800; $n_{other}$ = 150), how often the participant currently wears a mask in public ($n_{never}$ = 359; $n_{rarely}$ = 259; $n_{sometimes}$ = 337; $n_{often}$ = 197; $n_{always}$ = 167), and their political orientation (see S1 Appendix for visualizations of our data by political orientation).

## Study 2 Results

**Analysis notes.** All main text analyses were conducted on the full dataset. For each analysis, we only report the highest order interaction in the main text (which were explored, as planned, using Tukey's HSD). Complete analyses (including lower-order effects and subset analyses) are available in the S1 Appendix for interested readers. We also present detailed analyses of the Social Contribution and Social Acceptance scale in the S1 Appendix. However, because we anticipate that readers will be curious about the nature of effects for these scales, we present visualizations of our main analyses of these scales in Fig 2.

Finally, we report detailed analyses of the effects of both participant and target gender in the S1 Appendix. To summarize, as in Study 1, we found that masked targets were more likely to be assumed to be female ($B$ = -.04, $SE$ = .01, $p$ = .004). Unlike in Study 1, participants' masking behavior was not predicted by their own gender ($p$ = .85).

**Perceived classroom fit.** As in Study 1, we predicted Perceived Classroom Fit from the target's mask-wearing behavior, the class's mask-wearing behavior, and their interaction. Of importance, there was a significant interaction ($B$ = .22, $SE$ = .009, $p$ < .0001, $\eta_p^2$ = .296; Fig 3) that followed the same pattern as in Study 1: Masked conformers were perceived to have the highest classroom fit ($M$ = .37), and were rated as fitting in significantly better than both masked deviants ($M$ = .009, $p$ < .0001, $d$ = 1.08 [$CI_d$ = 93–1.23]) and unmasked deviants ($M$ = -.24, $p$ < .0001, $d$ = 1.80 [$CI_d$ = 1.64–1.96]). Unmasked conformers had the second highest perceived classroom fit scores ($M$ = .28); they were rated as fitting in better than both unmasked deviants ($M$ = -.24, $p$ < .0001, $d$ = 1.52 [$CI_d$ = 1.36–1.67]), and masked deviants ($M$ = .009, $p$ < .0001, $d$ = .798 [$CI_d$ = .65–.95]).

Finally, maskers were perceived as fitting in better than non-maskers: masked conformers had higher perceived classroom fit than unmasked conformers ($p$ = .002, $d$ = .28 [$CI_d$ = .13-.43]) and masked deviants had significantly higher fit than unmasked deviants ($p$ < .0001, $d$ = .72 [$CI_d$ = 58-.86]). Unplanned analyses reported in full in the S1 Appendix revealed that this effect was not driven by the participants' own masking behavior (S1 Appendix): these effects emerged whether the participant never/rarely/sometimes/often/always masked themselves, model fit was not improved by adding participant masking behavior, and our modal participant reported "never" masking.

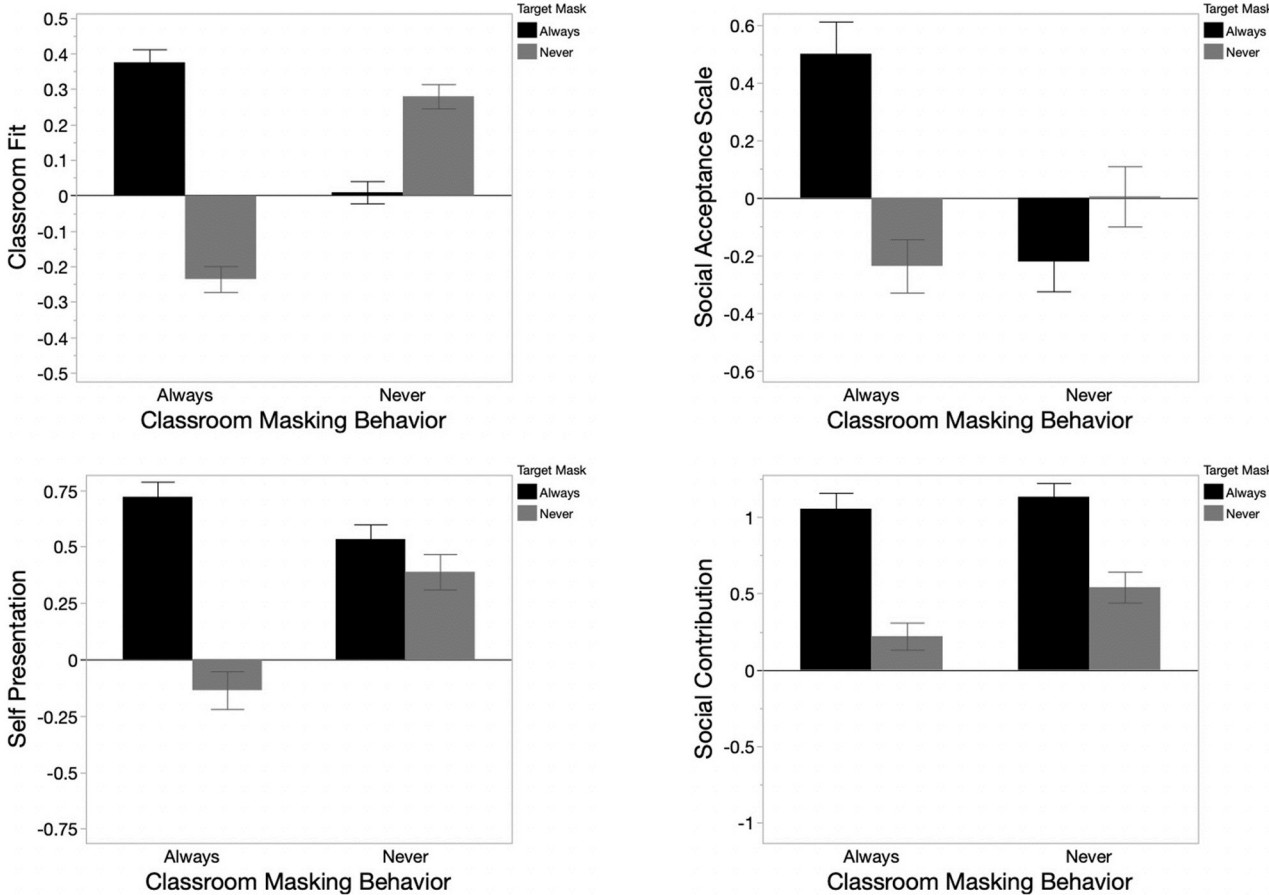

**Fig 2. DV (y-axis) by class masking behavior (x-axis), and target masking behavior (gray vs. black).** Error bars are 95% CIs. Going clockwise beginning in the upper left, DVs are: Perceived Classroom Fit (main text), Social Acceptance (S1 Appendix), Self-Presentation (main text), and Social Contribution (S1 Appendix).

We next tested whether knowledge of immune status mitigated the effects of mask conformity on judgments. To this end, we predicted perceived classroom fit from target masking behavior, classroom masking behavior, immune status, and their interactions. We found a significant three-way interaction ($p$ = .0006, $\eta_p^2$ = .0085; Fig 3). Planned Tukey tests revealed that this interaction was driven by unmasked conformers: unmasked conformers were perceived as fitting in significantly better when no immune-status information was provided ($M$ = .395) than when either the target ($M$ = .22, $p$ = .002, $d$ = .58 [CI$_d$ = .27, .89]) or professor ($M$ = .25, $p$ = .009, $d$ = .479 [CI$_d$ = .18, .78]) was known to be immunocompromised. However, immune-status made no difference in perceived classroom fit among masked conformers, unmasked deviants, or, critically, masked deviants (all $p$>.1).

**Secondary DVs.** We next predicted Self-Presentation, Social Acceptance, and Social Contribution from classroom masking behavior, target masking behavior, and their interaction (Fig 2). For all DVs, there was a two-way interaction of target and class-masking behavior (all $p$ < .01). Three-way interaction with immune-status never emerged; detailed and full reporting of all analysis, including lower-order effects of immune-status are in the S1 Appendix.

Because we used a modified version of the Self Presentation scale in Study 3, we describe results for the Self Presentation scale here. Recall that the Self Presentation scale captured the

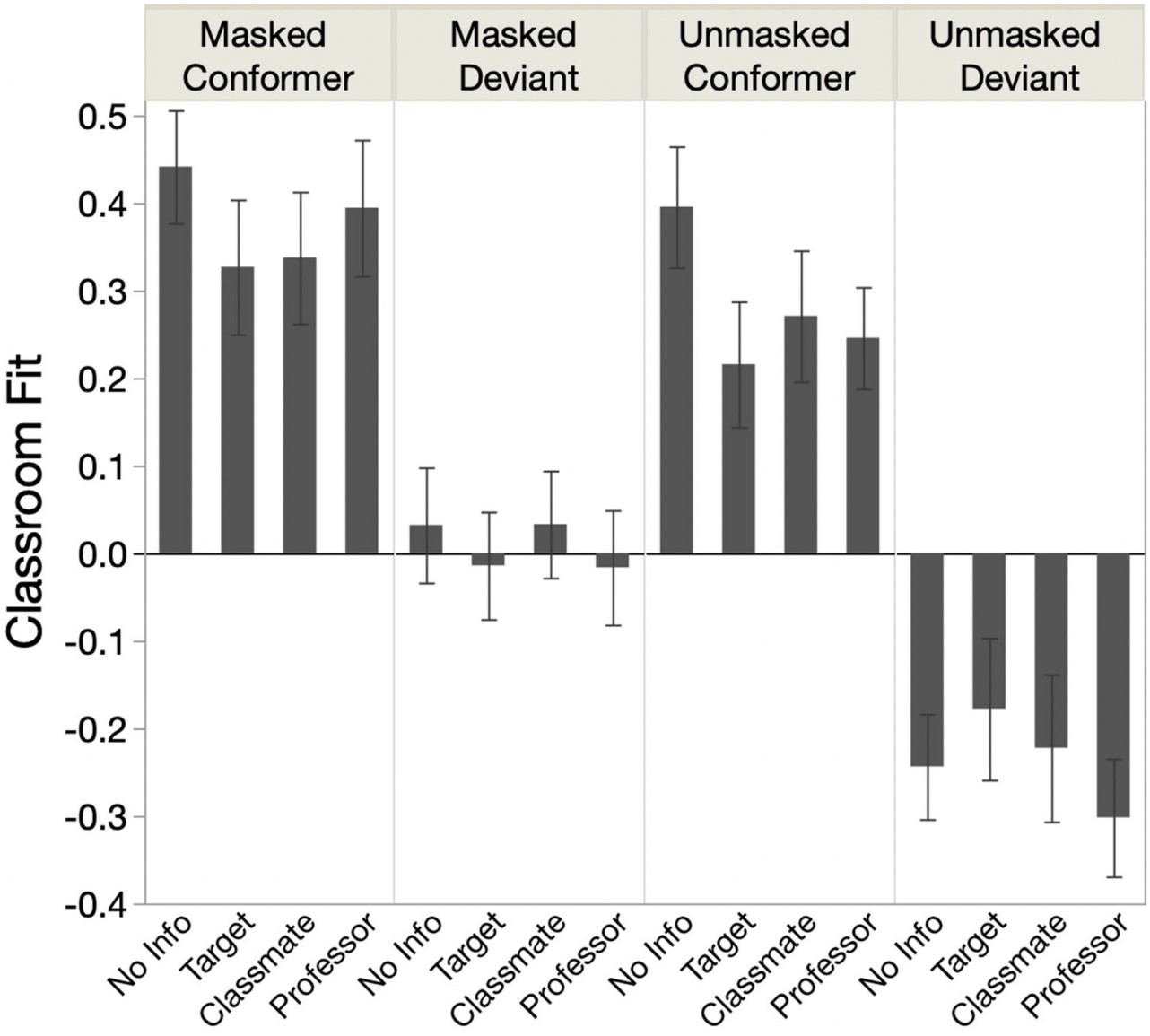

**Fig 3. Effect of immune status (x-axis) on perceived classroom fit (y-axis); error bars are 95% CIs.**

extent to which participants believed that B puts effort into how they are perceived by others (Table 2), a variable that is known to predict health behaviors [14, 41, 62], and that some have theorized might be related to the desire to mask [63]. We found that B's perceived level of concern for self-presentation differed for each masking cell (S1 Appendix). Masked conformers had the highest self-presentation scores ($M = .72$) and unmasked deviants had the lowest ($M = -.14$), and these cells differed significantly from one another ($p < .0001$, $d = 1.23$, [CI$_d$ = .1.07, .1.39]). Indeed, Tukey's HSD revealed that masked conformers had significantly higher self-presentation scores than any other target type (all $p < .01$; all $d > .25$), and unmasked deviants had significantly lower self-presentation scores than any other target type (all $p < .0001$; all $d > .74$). Unlike for perceived classroom fit, both masking *and* masking conformity were strong cues to self-presentation. A consequence of this was, for example, that masked deviants ($M = .53$) had significantly higher self-presentation scores than unmasked conformers ($M = .39$; $p = $

.03, $d$ = .21 [CI$_d$ = .06, .36]; see Fig 2). These data suggest that individuals who masked were perceived as caring more about how others perceive them, and that individuals who conformed were also perceived as caring more about how others perceive them.

## Study 3 Discussion

Study 2 assessed the impact of mask-wearing and immune-status on perceptions of students. As in Study 1, deviance from dominant masking behavior led to the expectation that the target student would experience social and academic penalties within the classroom (perceived likelihood of being invited to a study group; being liked by the professor). Troublingly, knowing that someone in the classroom was immunocompromised rarely mitigated the effects of masking or deviance on expected social judgments. Even masked deviants (who wore masks when others did not) who were described as being immunocompromised were still expected to fit in with their classmates relatively poorly. These data raise concerns about how immunocompromised individuals may be particularly harmed by environments where mask-wearing is a personal choice instead of a public mandate [64].

Masking also impacted perceptions of the target's social contribution, social acceptance, and self-presentation. These three outcome measures–which are important within the health psychology literature [14, 41, 65, 66]–suggest that *both* masking *and* mask conformity might benefit one's perceived social image, perceived social capital, and perceived social well-being. However, the pattern of data for these three scales was not identical to the pattern of data for the Classroom Fit scale (see Fig 2). These effects argue against the possibility that our reported conformity effects on perceived Classroom Fit are explained by a simple 'conformers are expected to be perceived more favorably' heuristic. Similarly, it is unlikely that these effects can be explained by participants' tendencies to evaluate the target's masking behavior against their own: while participants rated maskers and mask-conformers more positively than non-maskers and deviants, the modal participant reported "never" masking when out in public, and participant-masking behavior did not predict any social judgments.

Although the experimental design of Studies 1 and 2 allows us to draw causal inferences about the impact of masking on perceptions of students, they leave open several questions that are addressed in Study 3. One critically important question is the extent to which these findings generalize to the actual college classroom context. Studies 1 and 2 focus on perceptions of a hypothetical target college student, not actual social/academic penalties. Of course, studies of beliefs—like Studies 1 and 2—are valuable because whether or not mask-wearing conformity *actually* leads to differences in the treatment of individuals, the *belief* that it does could cause individuals to self-select out of educational environments. Or, individuals might alter their mask-wearing decisions due to fears of experiencing social and professional penalties. Put differently, when considering classrooms where masking is rare, students who want or need to mask may feel that they must choose between protecting their health, building friendships/ connections, and advancing their careers. Study 3 was designed to more clearly understand the real-world strategies that students use to avoid being mask deviants.

## Study 3

"...*I took my mask off [in a context where others were masking]. Everyone looked at me weird and was mumbling under their breaths about it.... I ended up leaving the class mainly [']cause I was getting mad.*"

–Free Response, Study 3, Participant 99

Experiments 1 and 2 demonstrated that masking deviants were expected to be perceived less favorably than masked conformers. Do students actively avoid being mask deviants, and if so, how do they achieve that goal? On the one hand, students who believe their mask-wearing behavior will deviate from that of their classmates may avoid particular classes, thus constraining further professional opportunities. Or, as in the quote above, upon realizing that their behavior deviates from the classroom norm, they may remove themselves from the class entirely. On the other hand, students who are committed to attending particular classes may feel pressured to alter their masking behavior–regardless of their personal needs–in order to avoid perceived social and professional penalties. In Study 3, we address these possibilities by studying real college students' reports of the impact of mask conformity on their educational experience during and after the acute phase of the COVID-19 pandemic.

In addition, we built on Experiments 1 and 2 by collecting both qualitative responses and frequency data describing the actual experiences of college students as they navigate masking. Specifically, we modified many of the scales from Experiments 1 and 2 to a self-report format in order to assess the impact of masking from the student perspective. As with our other Studies, we also tested the impact of group membership on our outcomes of interest.

## Study 1 Method

**Testing context.**   Data were collected online in late April and early May, 2023 from participants located within the U.S.

**Participants.**   Most participants were recruited from CloudResearch panels [42], with a small subset coming from SONA at Skidmore College. This means that participants came from a diverse array of colleges. Our pre-registered target sample size was $N = 200$, and a total of $N = 233$ consented and met the inclusion criteria described above. Of these, $n = 50$ were excluded based on preregistered criteria (failing attention checks, failing to provide intelligible long-form results). This resulted in a final $N = 183$ who completed all portions of the study; of these, $n = 162$ provided responses to all questions and scales, with the remaining $n = 21$ providing data on a subset of scales.

## Materials and procedure

**Inclusion screening.**   Participants completed an inclusion screening described in the S1 Appendix to ensure that they met our minimum inclusion criteria. Eligible participants reported that (a) they have worn a mask at least once and (b) they were currently college students *and* had attended at least one in-person semester of college courses between Fall, 2020 and Spring, 2023 (inclusive).

**Free response.**   Eligible participants were randomly assigned to one of four conditions: they were asked to recall a time when they [were/were not] wearing a mask while most people in their environment [were/were not] wearing a mask. In other words, they were asked to discuss a time when they were either a masked conformer, an unmasked conformer, a masked deviant, or an unmasked deviant. They were asked to describe the situation, why they thought they were in the situation, how they felt about themselves, whether the experience impacted their future decisions, and whether they felt penalized for their behaviors (see OSF for full study materials). We asked that participants focus their response on classroom contexts when possible. Participants were instructed that if they had *not* been in the masking situation described, they should clearly say so, and should speculate about why they had not been in that situation.

As planned, these free responses were coded in several ways. First, we coded whether participants indicated that they had *never* been in that particular situation (reported in the Results

**Table 3. Each item in the self-presentation scale, alongside the percent of respondents who endorsed (i.e. responded above the midpoint) each item.** We pair these data with representative quotes from the Free Response task that illustrates each item. A '-' in the Free Response column indicates that we did not find any quotes that spoke directly to (or against) that item.

| I think my masking behavior impacts. . . | Percent above midpoint | Free Response |
|---|---|---|
| people's impressions of me | 57% | "i did not want people to think i was anti-mask or didn't believe in covid, and i also did not want to spread/catch. i did feel penalized. it made me remember my mask more often in order to not be in that situation." |
| whether others respect me | 45% | ". . .if i did not wear one I was looked down upon" |
| people's impressions of my community | 42% | "I wanted to confirm [sic] to social standards, and I wanted to portray myself as a certain way politically" |
| whether others like me | 39% | "People judged me for it the next day saying I don't wear masks and am trying to get people sick which annoyed me because I wasn't doing that at all I was just getting a drink. Yes I felt penalized because everyone was judging me for it" |
| how well off I appear | 33% | - |
| people's impressions of my family | 33% | - |
| whether people perceive me as a difficult student | 29% | "I did not think much of the other classmates, other than being happy to see they were willing to comply with wearing a mask, as opposed to making a fuss and disrupting things." |
| how smart I sound/look | 28% | "Even months after I was still wearing a mask and I would frown upon anyone that wasn't doing the same. In my eyes they were dumb and not trying to be safe." |
| people's impression of whether I have financial difficulties | 18% | - |

section below), allowing us to assess whether participants had actively avoided e.g., being deviants. Second, because Study 1 focused narrowly on STEM classrooms, we coded each response for whether it was explicitly stated to have occurred within a STEM classroom; these data are reported in the S1 Appendix because very few participants spontaneously mentioned whether they were (or weren't) talking about a STEM classroom (S1 Appendix). Finally, we coded whether each response was thematically relevant to our study (e.g., reference conformity/deviance, masking norms, penalties for masking, gender/race/immune status, or any of the themes that were measured in our quantitative scale below). These data are reported in Tables 3 and 4.

**Scales.** Participants completed the Mask Impact Scale (Tables 2 & 4), the Class Choice Scale (Table 2), and the Self Presentation Scale (Tables 2 & 4).

**Other demographics.** We asked participants' gender ($n_{male}$ = 54; $n_{female}$ = 100; $n_{other}$ = 7), how often the participant currently wears a mask in public ($n_{never}$ = 31; $n_{rarely}$ = 55; $n_{sometimes}$ = 39; $n_{often}$ = 30; $n_{always}$ = 6), their race/ethnicity ($n_{POC}$ = 83, $n_{white}$ = 78), and whether they identified as immunocompromised ($n_{immunocompromised}$ = 18 $n_{not}$ = 143). We also asked about participants' masking preferences (e.g., how often they WOULD wear a mask in public if all constraints were removed)—these data were so similar to participants' actual masking behaviors that we report them only in S1 Appendix.

## Results

**Analytic choices.** As planned, our analyses are primarily descriptive; we argued in our preregistration against focusing on measures of central tendency for this dataset (though full data are available for any individuals who wish to explore such measures), as we were primarily focused on understanding responses from the groups who have been most impacted by

Table 4. Each item in the mask impact scale, alongside the percent of respondents who endorsed (i.e. responded above the midpoint) each item. We pair these data with representative quotes from the Free Response task that illustrate each item. A '-' in the Free Response column indicates that we did not find any quotes that spoke directly to (or against) that item.

| I feel that… | Percent above midpoint | Free Response |
|---|---|---|
| Others like me more because of my masking behavior | 45% | - |
| I have to choose between my safety and fitting in when it comes to masking | 35% | "when I wear a mask when most of the people around me were not wearing a mask, I usually feel safer, but I also feel like a weirdo sometime." |
| All else being equal, I will change my masking behavior to fit in with others | 33% | "I have not been in this situation because I follow the trends of other people. I tend to do as others do." |
| My own masking preferences have limited my access to social interactions with fellow students | 29% | "those who chose not to wear them seemed to disassociate themselves from the rest of us. It seemed that firendships [sic] and teams were strained by these choices." |
| My classmates would be less likely to invite me to study group if I failed to conform to the dominant masking behavior in my class | 26% | - |
| My professor will like me less if my masking behavior differs from that of my classmates | 24% | "My professor claimed that masks limited the ability of people to connect so that taking masks off was a requirement to do well in the class. … She did not care what your reason was for wearing a mask because she considered covid a small threat." |
| The impact of my masking decisions is GREATER than it would be for others because of my race or gender | 20% | - |
| Others like me less because of my masking preferences | 19% | "…One day I lost my mask before class and decided to go in without one. I felt very self conscious and that others were looking at me as uncaring and self centered. Despite having a few others in the class with me that were not wearing masks I still felt it reflected poorly on my character and others were judging me." |
| I have felt ashamed for my masking choices in the classroom | 19% | "…it was after I had to make my walk of shame up to the front of the classroom and grab one of her disposable bright pink masks that I decided to always have a mask in my backpack, in my car, and for a few week after the instance first occurred, I even carried an extra mask on my key ring." |
| My own masking preferences have limited my access to academic resources | 15% | "…I took my mask off… Everyone looked at me weird and was mumbling under their breaths about it…I ended up leaving the class mainly cause I was getting mad. I should notnhave [sic] left class ['] cause there was a pop quize [sic] but people's actions made mendo [sic] so." |

COVID-19 and by masking. In describing the data, we focus largely on whether participants' responses fell above (general agreement) vs. at or below (neutrality and disagreement) the midpoint of each scale. In addition, as planned, while we lack the power to meaningfully compare data from structurally marginalized groups (women and non-binary individuals, people of color, and immunocompromised individuals) to structurally dominant groups (men, White people, and non-immunocompromised individuals), whenever possible, we aimed to center

data from marginalized groups. Full data are available on OSF for interested researchers who wish to explore these issues using other analytic approaches.

**Free response.** One goal of the free response section was to generate qualitative data and participant-generated language surrounding the impact of masking on the classroom experience. Full quotes are available in our dataset posted on OSF; selected quotes are presented throughout the manuscript, and are available in Tables 3 and 4 below to demonstrate the connection between participants' scale responses and their qualitative reports.

Studies 1 and 2 showed that those who deviated from dominant masking behavior were perceived as fitting in less well than those who conformed. Thus, our secondary goal was to assess whether participants had taken active steps to avoid deviating from the masking norm. Recall that participants were randomly assigned to provide responses about a time when they had been a masked conformer (i.e. wore a mask when others did; $N = 44$), an unmasked conformer (i.e. didn't wear a mask, and neither did anyone else; $N = 50$), a masked deviant (i.e. wore a mask when others did not; $N = 45$), or an unmasked deviant (i.e. failed to wear a mask when others did; $N = 43$). We measured whether participants reported that "I have never been in this situation," in order to explore whether participants had potentially avoided situations in which they would deviate from the norm in their environment.

Critically, participants were more likely to report having engaged in conforming behavior than in deviant behavior. Specifically, less than 10% of participants reported that they had *never* been a masked conformer, while 32% reported that they had *never* been an unmasked conformer. In other words, the majority of the sample reported having conformed. In contrast, 56% reported that they had never been a masked deviant, and 65% reported that they had never been an unmasked deviant (Fig 4). In other words, only a minority of the sample reported ever having deviated from masking norms. While Studies 1 and 2 demonstrated that deviants were expected to be perceived less favorably than conformers, Study 3 amplified these findings by demonstrating that individuals were less likely to report ever having been a deviant than they were to report ever having been a conformer. This, in turn, raises a question which we address in the analyses below: How do individuals avoid being deviants? Do they do so *only* by changing masking behavior (i.e. altering their health risk level) or *also* by changing which academic courses they are willing to attend (i.e. altering their academic trajectory)?.

**Class enrollment.** We next asked whether masking behavior impacts students' willingness to enroll in or remain in a course using the Class Choice scale (Table 2). After all, one way to avoid deviant behavior is to opt into environments in which *your* preferred masking behavior matches the dominant masking behavior. Consistent with this, overall, 60% of participants endorsed the idea–for at least one of the three questions asked–that masking behavior would impact how much they would want to enroll in or stay in a course (see S1 Appendix for full reporting). As preregistered, our key question was about participants' endorsement of the item "I am more likely to want to enroll in/stay in a class if I know that the typical masking behavior in the class will match my own". Overall, 48% of participants endorsed this statement (i.e. provided a response above the midpoint), with only 2% of participants selecting 'strongly disagree'. These data suggest that as of May, 2023, students still considered masking behavior–and in particular mask conformity–when making educational decisions. Planned *post-hoc* analyses revealed that 61% of immunocompromised individuals endorsed this item, compared to 43% of non-immunocompromised individuals. Due to limited sample sizes for immunocompromised individuals ($n = 18$, or 11% of participants), we avoid calculating statistical comparisons for this group relative to others, but instead simply note that these findings are suggestive of the possibility that immunocompromised individuals (who may not have the luxury of altering masking behavior) are especially likely to need to consider the dominant masking behavior when making academic choices about course enrollment (Table 5).

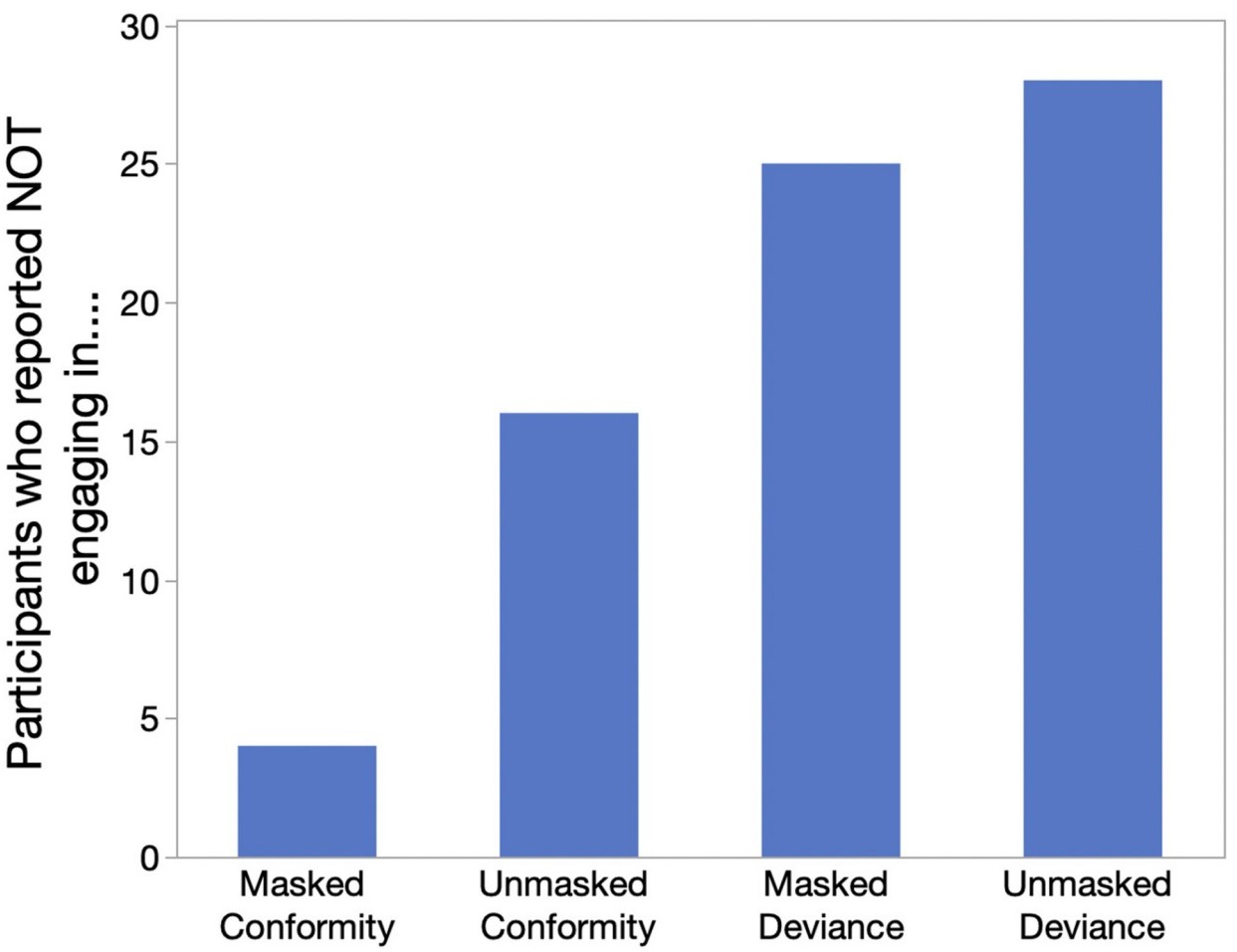

**Fig 4. Number of individuals reporting that "I have never been in this situation".**

**Self-presentation.** We next considered the extent to which participants indicated that masking impacted how favorably others perceived them. Overall, 77% of participants provided at least one above-midpoint response (i.e. agreed with the statement), with 18%–57% of participants endorsing each item (Table 3). These data suggest that a meaningful subset of college students believe that masking impacts how others perceive them within the classroom. In Table 3, we also report participant quotes from the free response section of the study that speak to each item within the Self-Presentation scale.

**Mask impact.** We next considered responses on our Mask Impact scale (Table 4). Once again, the vast majority of respondents (85%) endorsed at least one item on the scale. Consistent with Studies 1 and 2, we found evidence that a substantial subset of our sample perceived

**Table 5. Effect size and confidence intervals for the effect of masking on our three main DVs for Study 3.**

| | POC—White | Men—Women/Non-binary | Immunocompromised—not |
|---|---|---|---|
| **Classroom Choice (average)** | $d = .18$ [$CI_d = -.13, .49$] | $d = .32$ [$CI_d = -.02, .65$] | $d = .28$ [$CI_d = -.22, .77$] |
| **Mask impact (average)** | $d = .23$ [$CI_d = -.08, .54$] | $d = .20$ [$CI_d = -.14, .53$] | $d = .43$ [$CI_d = -.06, .93$] |
| **Self Presentation (average)** | $d = .06$ [$CI_d = -.25, .37$] | $d = .22$ [$CI_d = -.11, .56$] | $d = .70$ [$CI_d = .20, 1.19$] |

that they had experienced social, educational, and health penalties due to their masking behaviors.

**Group membership.** As expected, we did not have sufficient power to identify whether there were statistical differences in scores on our three main measures across demographic groups, and so we did not conduct statistical analyses on these data. However, across each of our three measures, mean level of endorsement of masking impact was higher for People of Color than for White people, higher for men than for women/non-binary respondents, and higher for immunocompromised individuals relative to non-immunocompromised individuals (see Table 5 for estimated effect sizes). These data suggest that among actual students, the perceived impact of masking may differ depending on group membership.

## General discussion

*". . .I felt a bit embarrassed to be one of the only ones wearing a mask, because I don't like to stand out. I felt that it made me more noticeable. The professor eventually asked everyone to continue wearing masks, which then made me feel a lot more comfortable."*

-Free Response, Study 3, Participant 168

While the acute phase of the COVID-19 pandemic shifted after vaccines became broadly available in early 2021, subsequent waves and variants of this and other illnesses suggest that decisions about (not) wearing face masks will likely remain a reality of public life. In the present study, we asked about the impact of masking on social and academic judgments of students within the college classroom. In Experiments 1–2, we manipulated (a) a hypothetical student's masking behavior, (b) the masking behavior of their classmates, and (c) their group membership (e.g., gender and immune status). We then measured the impact of this manipulation on perceptions of that student's fit within the classroom. In Study 3, we asked real college students from across the U.S. about their own masking behavior, their own group membership, and the perceived and experienced consequences of that behavior within their actual classrooms.

Across all three studies, we found two consistent effects. The first was that maskers were generally expected to be perceived more positively than non-maskers, even several years after the acute phase of the COVID-19 pandemic, and even among participants who themselves rarely masked. However, this main effect of masking was eclipsed by a much stronger effect of masking *conformity*. This manifested in the belief that conformers (regardless of actual masking behavior) were expected to fit in better than deviants within the classroom (Studies 1 and 2), that conformers were perceived as being more attentive to self-presentation concerns (Studies 2 and 3), that real students expressed actively avoiding situations of mask non-conformity (Study 3), and that real students expressed experiencing social and academic penalties from deviating from masking norms (Study 3). In other words, adherence to group norms around masking behavior (over and above the behavior itself) seemed to shape important social and educational judgments, even in a context with life-or-death implications for both individual and community health. We argue that it may be particularly important to better understand the extent to which health-related behaviors that are typically thought of as freely-chosen (i.e., masking in the post-mandate age) may indeed be constrained by conformity pressures.

Of importance, our data suggest that students avoided deviating from masking norms. How did they achieve this goal? Both our quantitative and qualitative data within Study 3 demonstrate that students took multiple approaches to avoid deviance. A plurality of students reported that their decision to enroll in or remain in a class was impacted by whether the

dominant masking behavior in the class matched their own masking preferences. These data are troubling, in that they suggest that mask conformity pressures could impact students' curricular choices, including serving as an obstacle to acquiring important skills and information. It stands to reason that if a required class for a major also requires students to choose between fitting in and their own health, students may consider altering their academic trajectory to avoid wrestling with this difficult choice several times a week.

In addition to altering their course enrollment decisions, many students reported altering their masking behavior to match that of their classmates. However, as highlighted by both Studies 2 and 3, immunocompromised individuals (and those who interact frequently with individuals who are especially vulnerable to COVID-19) may not have the option of safely changing their masking behavior to conform to social norms. And, more generally, as colleges, universities, and individual professors move away from mask mandates and towards a model of 'personal choice," the fact that individuals report changing their masking behavior to fit the visible majority could lead to a scenario where students who must remain in a primarily non-masking classroom feel pressure to remove their masks. Indeed, 35% of our sample in Study 3 indicated that they felt like they were choosing between their safety and fitting in within the classroom.

These data provide a novel demonstration that classic conformity effects apply to the context of face-masking, and add to an existing body of work suggesting that adherence to social pressures and self-presentational concerns can undermine individual health behaviors [51]. This matters because while even public-health experts like CDC leader Greta Massetti frame face-masking as an individual choice (e.g., conveying that "people may choose to wear a mask at any time, based on personal preference" [67]), the social and academic consequences of conforming to or deviating from dominant masking behavior renders it anything but. We encourage students, educators, and policy-makers (both within academic institutions and more generally) to consider how norm-setting surrounding masking may shape not only the spread of airborne disease, but also professional and social outcomes for people within different communities.

Of course, not all groups are equally impacted by COVID-19, or by the social and academic penalties that might emerge from failure to conform. We iteratively explored the impact of people's group membership (e.g., gender, immune-status, race) on perceptions of individuals in the classroom as a function of their masking behavior. Given that gendered expectations surrounding masculinity can impact rates of risky health behaviors [51], all three studies explored whether gender impacted the effects of masking or masking conformity on social and academic judgments. In Studies 1 and 2, we found mixed evidence for an interaction of gender and masking. Masked individuals were generally assumed to be female, suggesting that participants perceive an association between gender and masking behavior. While we did not find that gender interacted with conformity pressures (i.e. both men and women were penalized equivalently for deviating from masking norms), Study 3 suggested that men perceived themselves to experience greater impacts of masking non-conformity. These data are consistent with reports that some men avoided masking during the peak of the COVID-19 pandemic due to a desire to display masculinity [68].

Given that one of the goals of public health is to protect the community at large, public health decisions must center the needs of individuals who are already marginalized within and by the healthcare system. While much of the discourse surrounding masking policies has focused on reducing disease spread, social and emotional well-being is also a critical component of health [69]. We show that those in the masking-minority may incur social and academic penalties [33], suggesting a possible social health benefit to mask mandates. Within the context of the present study, we found that participants expected mask deviants (relative to

conformers) to be less likely to be invited to study group or to be liked by their peers and professor (Studies 1 and 2); Study 3 showed that a sizeable subset of participants perceived these effects of mask conformity on classroom fit to be their lived reality at college. Critically, these penalties were not mitigated by the knowledge that immunocompromised individuals were present in the classroom, suggesting that the absence of mask policies may disproportionately impact individuals who most need to wear masks. Indeed, while our sample size of immuno-compromised participants in Study 3 was small, they reported much higher impacts of masking pressure on social and academic outcomes than any other group, across all measures.

## Limitations and future directions

The current work afforded both high experimental control (Studies 1 and 2) for examining the causal impact of masking and masking conformity on judgments of students, and also ecological validity (Study 3) in demonstrating the existence of real-world consequences in students' lives. However, our approach was not without limitations.

**Limitations to participant recruitment.** Our research was conducted online, and was constrained to a particular time (2022–2023), place (the United States), language (English) and sample (predominantly participants from Cloud Research). Online data collection methods can quickly generate large, relatively diverse samples. However, additional work should explore the extent to which our results generalize beyond the current context. Specifically, collecting targeted sampling from individuals from particular groups (e.g., immunocompromised individuals, individuals from minoritized racial and gender groups, and individuals with extreme masking views) would likely improve the generalizability of these findings. Future work could also explore whether the current results are moderated by additional relevant participant demographic variables (e.g., socio-economic status, geography), could focus more explicitly on the roles of race and ethnicity in shaping the perceptions of masking behavior, and could study how these effects change over time in response to changing levels of disease risk.

**Limitations to study design.** In order to test individuals' perceptions of masking behavior, we also needed to develop new scales (e.g., the Classroom Fit scale), meaning that some of the scales in our dataset had not been previously normed. While we iterated these scales across the three presented studies, we acknowledge that the creation of new means of measurement to study new phenomena (e.g., face-masking in response to COVID-19) has inherent limitations. Future work should seek to replicate and extend our findings using the current and additional measures, while also providing additional psychometric information about the properties of our novel scales. Additionally, future research could fruitfully examine the consequences of masking and masking conformity on other social and professional outcomes and in additional contexts, including the workplace, informal social settings, and K-12 education.

**Other limitations to study interpretation.** Since this research was focused on the college classroom context, we do not know the extent to which these conformity pressures extend beyond classroom walls. This is especially important because college classrooms are spaces that individuals choose to attend (e.g., in whether they attend college, which classes they attend), even if those choices can have high-stakes consequences or be constrained by other factors (e.g., socioeconomic status). However, there are many settings where individuals do *not* have a choice about whether to be in a particular space (e.g., elementary school classrooms, health-related settings). In those contexts, individuals may feel particular pressure to alter their masking behavior to conform with masking norms. Previous work has shown that self-presentation concerns weigh strongly on individuals' minds, particularly those who are minoritized backgrounds, when seeking even routine health care [70, 71]. Again, if individuals feel that

they may be judged for masking (or not masking) in healthcare settings, this could exacerbate barriers to public health goals. Future research should expand upon the foundation provided by this work by studying the impacts of mask conformity pressures on contexts outside the college classroom.

Additionally, the current work did not directly test the specific cognitive mechanisms accounting for the impact of masking conformity on perceptions of target students, or the subsequent behaviors of real-world students. While our data are consistent with broad theoretical accounts related to conformity [24], norm-following [72], and self-presentation concerns [40, 73], future work will benefit from taking a more rigorous mechanistic approach to understanding the processes underlying the behaviors described within the present study. Critically, we have made all of our data available–including participants' qualitative responses, and we hope that our data will provide a strong starting point for researchers interested in doing theory-building.

## Conclusions

These data raise interesting theoretical questions at the intersection of health and social psychology. By highlighting a critical role for conformity in shaping judgments about students' mask-wearing behavior, we demonstrate that one highly visible public health choice (masking vs. not) may provide an important cue to group membership. These data are a starting place for asking about the causal impact of mask-wearing behavior on academic, professional, and social belonging. In addition, building on a long tradition of studying the role of attire in signaling social group membership [74], these data suggest that mask-wearing behavior may provide a strong cue to group belonging, much as other non-public-health-related visual signals (e.g., athletic jerseys). In sum, conformity demands may disproportionately pressure some people to choose between their well-being and their social and professional success. This raises concerns that the shadow of the COVID-19 pandemic may stretch far into the future, impacting both health and occupational development.

## Supporting information

**S1 Appendix. All supporting information is available in S1 Appendix.**
(DOCX)

## Author Contributions

**Conceptualization:** Jessica Sullivan, Corinne Moss-Racusin, Kengthsagn Louis.

**Data curation:** Jessica Sullivan, Corinne Moss-Racusin, Kengthsagn Louis.

**Formal analysis:** Jessica Sullivan.

**Funding acquisition:** Jessica Sullivan.

**Investigation:** Jessica Sullivan, Corinne Moss-Racusin, Kengthsagn Louis.

**Methodology:** Jessica Sullivan, Corinne Moss-Racusin, Kengthsagn Louis.

**Project administration:** Jessica Sullivan.

**Resources:** Jessica Sullivan.

**Validation:** Jessica Sullivan.

**Visualization:** Jessica Sullivan.

**Writing – original draft:** Jessica Sullivan, Corinne Moss-Racusin, Kengthsagn Louis.

**Writing – review & editing:** Jessica Sullivan, Corinne Moss-Racusin, Kengthsagn Louis.

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
