## [Decision Letter · Decision Letter 0]

22 Apr 2024

PONE-D-24-06722Masking is good, but conforming is better: The consequences of masking non-conformity within the college classroomPLOS ONE

Dear Dr. Sullivan,

Thank you for submitting your manuscript to PLOS ONE. After careful consideration, we feel that it has merit but does not fully meet PLOS ONE’s publication criteria as it currently stands. Therefore, we invite you to submit a revised version of the manuscript that addresses the points raised during the review process.

Please include the following items when submitting your revised manuscript:A rebuttal letter that responds to each point raised by the academic editor and reviewer(s). You should upload this letter as a separate file labeled 'Response to Reviewers'.A marked-up copy of your manuscript that highlights changes made to the original version. You should upload this as a separate file labeled 'Revised Manuscript with Track Changes'.An unmarked version of your revised paper without tracked changes. You should upload this as a separate file labeled 'Manuscript'.

We look forward to receiving your revised manuscript.

Kind regards,

Mary Diane Clark, PhD

Academic Editor

PLOS ONE

Journal Requirements:

"Funding from Skidmore College was awarded to JS in support of this project. "

5. Please include a copy of Table 3 which you refer to in your text on page 36.

Additional Editor Comments:

Thank you for your submission. I now have two reviews and both individuals have made comments on how to make the manuscript clearer. Please respond to their comments and return the manuscript by May 20, 2024. As written the manuscript is confusing to the reviewers.

Reviewers' comments:

Reviewer's Responses to Questions

**Comments to the Author**

1. Is the manuscript technically sound, and do the data support the conclusions?

Reviewer #1: Partly

Reviewer #2: Partly

2. Has the statistical analysis been performed appropriately and rigorously? 

Reviewer #1: I Don't Know

Reviewer #2: I Don't Know

3. Have the authors made all data underlying the findings in their manuscript fully available?

Reviewer #1: Yes

Reviewer #2: Yes

4. Is the manuscript presented in an intelligible fashion and written in standard English?

Reviewer #1: No

Reviewer #2: Yes

5. Review Comments to the Author

Reviewer #1: Thank you for the opportunity to review this manuscript.

General:

Timely and interesting manuscript about further exploration of masking behaviors using social and psychological theories. The data is collected from the study participants to give their responses about a hypothetical setting in their own classrooms and results are interpreted based on several “hypotheses” about social and psychological factors. The manuscript presentation is complex because the authors have decided to report three interrelated questions as three separated studies all combined in one manuscript. The difficulty for the audience is the poor delineation and clarity in background, assumptions, hypothetical situations, objectives, and methods for all the three studies. The details (especially background studies, rational and methods) seems mixed up, fragmented, and repetitive in different sections. It seems like there were initially three manuscripts which authors decided to submit as one without making necessary changes. Even though the premise of these studies is interesting, the reader may feel confused or bored (or both) the way this manuscript(s) is written, and results are presented.

Specific Comments:

Useful theoretical information in the introduction. Framing of research questions is appropriate. Rationale of choosing college classrooms setting and broad range of participants is provided. However, the background information, related studies and rationale is incomplete, and it seems that it is provided in different sections later.

The Current Research provides overall description of research studies mixed with some methods details. This section does not clearly delineate the distinguishing factors between these studies except studies #1 and 2 were experimental and were aimed at gathering “evidence that real-world masking behavior can be associated with the types of social and academic consequences”, while in study 3# “participants were asked to report their own experiences of other’s responses to their masking choices”.

There are links to the Supplemental online materials in two different locations. Reading the pdf. files in OSF link is crucial, because the assumptions, methods, and outcomes measures are described with more clarity than the manuscript itself.

Even though the details are there, this section seems fragmented and in the presence of individual study objectives, methods, and details, also seems redundant. Therefore, it will be useful for the reader if methods can be written in a more concise and clearer manner, with some sample questions. Our recommendation would be to provide all the background information and rationale in the beginning of the manuscript, followed by a consolidated method, results and discussion sections.

It is not clear whether study participants continuously observed their classrooms over a pre-specified time-period, or they responded based on a one-time observation. (Sorry if this detail was already provided because it is hard to keep up with details spread all over the paper and external supplemental files) Furthermore, all the responses are binary (always wears a mask, never wears a mask) which seems to be somewhat unrealistic. There could be instances where the target (B) chooses to wear a mask for some days when they were not well, but otherwise not 100% adherent. Same is true for the rest of the class or professor. If the authors have a better explanation for choosing a binary in this hypothetical situation, they should justify it with evidence.

The graphs are easy to follow, and the findings are interesting. However, there is hardly anything that surprises us. People like to conform, whether the prevalent culture is a positive or negative behavior. This happens for a fear of perceived social benefits which may overshadow health risks to self or others. Gender effects are also well documented in several studies.

So, making a culture of masking in the classroom will have the most impact on individuals’ behavior and their perception of being fit for the classroom environment.

Please add a description of limitations of this study in terms of study assumptions, study design, and participant recruitment.

Reviewer #2: Thank you for the opportunity to review this manuscritp. The authors present a series of three studies aimed at exploring the impact of masking and not masking on social and academic judgments of students within the college classroom. The series has coherence and use mixed-methods to have a wide picture of the phenomenon. This reviewer appreciates the authors' effort to collect such an amount of data on a topic that nowadays is fully pertinent.

Unfortunately, the way this series is framed in the literature and the data collection strategy reduce my enthusiasm for the manuscript.

Regarding the first concern, I have missed, both in the Introduction and Discussion, more reference to the vast literature on peer pressure, social acceptance, majority and minority influence, and related topics. I understand the irruption of COVID-19 has motivated research on topics that look new but that, indeed, are not. Articles on masking are dated from 2020. But masking is just another example of behavior that is subjected to social pressure, same as alcohol use and abuse, gender-consistent behaviors, and the like. As masking is not a "new" behavior, IMHO research on masking cannot focus on showing a picture of the phenomenon but should also explain such phenomenos according to one of more of the theories that have tradicionally addressed social pressure and majority and minority influence.

Regarding my second concern, authors are heavily relying on data collected with ad-hoc measures consisting on a few items. They report the alpha, that is good, but sometimes it is barely acceptable (around .70). The fact that data are not collected through validated measures may pose a problem and should be, at the least, acknowledged as a limitation.

6. PLOS authors have the option to publish the peer review history of their article (what does this mean?). If published, this will include your full peer review and any attached files.

Reviewer #1: No

Reviewer #2: No

---

## [Author Response · Author response to Decision Letter 0]

14 Jun 2024

Full Response to Reviewers is included in our cover letter.

---

## [Decision Letter · Decision Letter 1]

19 Aug 2024

PONE-D-24-06722R1Masking is good, but conforming is better: The consequences of masking non-conformity within the college classroomPLOS ONE

Dear Dr. Sullivan,

Thank you for submitting your manuscript to PLOS ONE. After careful consideration, we feel that it has merit but does not fully meet PLOS ONE’s publication criteria as it currently stands. Therefore, we invite you to submit a revised version of the manuscript that addresses the points raised during the review process.

Reviewer 2 from the first round of reviewers has given the paper an accept and I agree that it is much improved. I have a few more issues to correct some copyright type issues and then some questions about your analysis. I hope you can respond to some of these within the manuscript. 

We look forward to receiving your revised manuscript.

Kind regards,

Mary Diane Clark, PhD

Academic Editor

PLOS ONE

Journal Requirements:

Additional Editor Comments:

Thank you for fixing many of the organizational issues in the rewrite. I have a few issues related to editing and then some questions about the analysis.

On page 4 the 2nd paragraph, the 4th line says "from earlier DURING the COVID". please add during

Pn age 7 --You wrote across three Studies---please change it to

across three studies

page 31--didn't please change to did not for formal papers

You have extremely high power as can be seen in your effect sizes in study 1 and 2. Most of the analysis are regression but there are some that it seems like you need Bonferroni corrections. For example on page 22---it seems that these should have a higher p value given the number of analysis.

Then on page 30 you have p<.05 and d>.02. This section is clearly in contrast to the other high effect sizes and you do not mention that this MAY be related to the high N

Finally references are in Vancouver with numbers in text.

I look forward to your response to these issues and the manuscript will contribute to our public health literature.

Reviewers' comments:

Reviewer's Responses to Questions

**Comments to the Author**

1. If the authors have adequately addressed your comments raised in a previous round of review and you feel that this manuscript is now acceptable for publication, you may indicate that here to bypass the “Comments to the Author” section, enter your conflict of interest statement in the “Confidential to Editor” section, and submit your "Accept" recommendation.

Reviewer #2: All comments have been addressed

2. Is the manuscript technically sound, and do the data support the conclusions?

Reviewer #2: (No Response)

3. Has the statistical analysis been performed appropriately and rigorously? 

Reviewer #2: (No Response)

4. Have the authors made all data underlying the findings in their manuscript fully available?

Reviewer #2: (No Response)

5. Is the manuscript presented in an intelligible fashion and written in standard English?

Reviewer #2: (No Response)

6. Review Comments to the Author

Reviewer #2: (No Response)

7. PLOS authors have the option to publish the peer review history of their article (what does this mean?). If published, this will include your full peer review and any attached files.

Reviewer #2: No

---

## [Editor Report · Decision Letter 2]

7 Oct 2024

Masking is good, but conforming is better: The consequences of masking non-conformity within the college classroom

PONE-D-24-06722R2

Dear Dr. Sullivan,

We’re pleased to inform you that your manuscript has been judged scientifically suitable for publication and will be formally accepted for publication once it meets all outstanding technical requirements.

Kind regards,

Mary Diane Clark, PhD

Academic Editor

PLOS ONE

Additional Editor Comments (optional):

Thank you for correcting these final issues.
---

## [Editor Report · Acceptance letter]

12 Nov 2024

PONE-D-24-06722R2 

PLOS ONE

Dear Dr. Sullivan, 

I'm pleased to inform you that your manuscript has been deemed suitable for publication in PLOS ONE. Congratulations! Your manuscript is now being handed over to our production team.

Kind regards, 

on behalf of

Dr. Mary Diane Clark 

Academic Editor

PLOS ONE